# Mutational Patterns in Colorectal Cancer: Do PDX Models Retain the Heterogeneity of the Original Tumor?

**DOI:** 10.3390/ijms26115111

**Published:** 2025-05-26

**Authors:** Maria El Hage, Zhaoran Su, Michael Linnebacher

**Affiliations:** Molecular Oncology and Immunotherapy, Clinic of General Surgery, Rostock University Medical Center, 18057 Rostock, Germany; maria.elhage11@gmail.com (M.E.H.); zhaoran.su@med.uni-rostock.de (Z.S.)

**Keywords:** colorectal cancer, patient-derived xenograft, mutational patterns, tumor heterogeneity, preclinical models

## Abstract

Colorectal cancer (CRC) remains a leading cause of cancer-related mortality worldwide, highlighting the need for a deeper understanding of the genetic mechanisms driving its development and progression. Identifying genetic mutations that affect key molecular pathways is crucial for advancing CRC diagnosis, prognosis, and treatment. Patient-derived xenograft (PDX) models are essential tools in precision medicine and preclinical research, aiding in the development of personalized therapeutic strategies. In this study, a comparative analysis was conducted on the most frequently mutated genes—*APC*, *TP53*, *KRAS*, *BRAF*, *NRAS*, and *ERBB2*—using data from publicly available databases (*n* = 7894) and models from University Medicine Rostock (*n* = 139). The aim of this study was to evaluate the accuracy of these models in reflecting the mutational landscape observed in patient-derived samples, with a focus on both individual mutations and co-occurring mutational patterns. Our comparative analysis demonstrated that while the ranking of individual mutations remained consistent, their overall frequencies were slightly lower in the PDX models. Interestingly, we observed a notably higher prevalence of *BRAF* mutations in the PDX cohort. When examining co-occurring mutations, *TP53* and *APC* mutations—both individually and in combination with other alterations—were the most frequent in both datasets. While the PDX models showed a greater prevalence of single mutations and a slightly higher proportion of tumors without detectable mutations compared to the public dataset, these findings present valuable insights into CRC’s mutational landscape. The discrepancies highlight important considerations, such as selective engraftment bias favoring more aggressive tumors, differences in sample size between the two cohorts, and potential bottleneck effects during PDX engraftment. Understanding these factors can help refine the use of PDX models in CRC research, enhancing their potential for more accurate and relevant applications in precision oncology.

## 1. Introduction

Colorectal cancer (CRC) remains a leading cause of cancer-related deaths worldwide, underscoring the critical need to understand the genetic factors driving its development and progression. Identifying genetic mutations that disrupt key molecular pathways is vital for improving the diagnosis, prognosis, and treatment strategies for CRC patients [1]. Extensive research has shown that CRC develops through a multistep process influenced by many factors, where genetic mutations that disrupt key molecular signaling pathways are critical in both the initiation and progression of CRC [2,3]. Mutations in tumor suppressor genes have been recognized as key drivers of CRC, with more than 80% of CRC cases showing recurrent mutations in these genes [4,5,6]. *TP53* and *APC* mutations are considered crucial driver mutations in CRC [7]. In addition, recent clinical guidelines highlight *KRAS*, *BRAF*, *NRAS,* and *ERBB2* as key genes used to evaluate clinical treatment options in CRC [8]. Advances in genetic sequencing have revolutionized the field by enabling the rapid, comprehensive analysis of many genes and samples. This progress has improved the diagnosis, prognosis, and prediction of treatment responses for CRC patients, paving the way for personalized treatment strategies [9,10].

Patient-derived xenografts (PDX) are essential in precision medicine and preclinical research, enabling the development of personalized treatment strategies. The construction and different applications of PDX models is highlighted in Figure 1 [11]. While PDX models are valuable tools in the field of cancer research, it is important to acknowledge their limitations. A significant drawback of this approach is the progressive replacement of the human tumor stroma by murine-derived stromal cells. This alteration of the tumor microenvironment may have a substantial impact on therapeutic responses [12]. Furthermore, disparities in drug metabolism and molecular signaling pathways between mice and humans can limit the translational relevance of findings derived from PDX models [12]. The vasculature that develops within PDX tumors is of murine origin, potentially influencing tumor perfusion and the efficacy of drug delivery. Furthermore, the interactions between human tumor cells and surrounding murine tissues do not fully recapitulate the complexity of human tumor–host dynamics [13]. Finally, the variability in mouse strains and experimental protocols across different research centers contributes to limited reproducibility and consistency in experimental outcomes [13].

In this study, we conducted a comparative analysis of the most significant CRC mutations—*APC*, *TP53*, *KRAS*, *BRAF*, *NRAS*, and *ERBB2*—between publicly available databases, subsequently referred to as the clinical cohort, and PDX models, referred to as the PDX cohort, from our institution. We then examined the co-occurrence of these mutations. Our analysis addresses two main questions: First, we investigate whether PDX models accurately reflect the mutational landscape observed in CRC tumors. Specifically, we assess whether key driver mutations—*APC*, *TP53*, *KRAS*, *BRAF*, *NRAS*, and *ERBB2*—both individually and in combination, are preserved in the PDX cohort at frequencies comparable to those in a large clinical dataset. This comparison aims to determine whether PDX models maintain the same level of genetic complexity and heterogeneity found in patient-derived samples, despite potential selective pressure during engraftment. The second question explores whether PDX models can effectively capture the diverse and clinically relevant subgroups of CRC, as well as the broader heterogeneity observed across patient populations—an essential factor for their application in personalized therapeutic research and preclinical studies.

## 2. Results

### 2.1. Single-Mutation Frequency Comparison Between the Clinical Cohort and the PDX Cohort

The initial step entailed a comparative analysis of key mutations (*APC*, *TP53*, *KRAS*, *BRAF*, *NRAS*, and *ERBB2*) between the clinical cohort (*n* = 7936 cases), which is the collection of the public database cases, and the PDX model-derived cohort (*n* = 137) which is the collection of our HROC—short for Hansestadt Rostock CRC—models. In the clinical cohort, arranged in decreasing order from the most frequent to the least frequent mutation, *APC* was mutated in 67.6% of cases, followed by *TP53* at 65.3%, *KRAS* at 42.2%, *BRAF* at 9.5%, *NRAS* at 4.4%, and *ERBB2* at 2.5%. In the PDX cohort, mutation frequencies of 43.1% for *APC,* 42.3% for *TP53*, 20.4% for *KRAS* and *BRAF*, 2.2% for *NRAS*, and 1.5% for *ERBB2* were observed.

We then compared the frequencies of common oncogenic mutations between the clinical cohort and the PDX cohort using chi-square and Fisher’s exact tests, as appropriate. Significant differences were observed for *APC* (67.6% vs. 43.1%, χ^2^ = 35.72, *p* < 0.001), *TP53* (65.3% vs. 42.3%, χ^2^ = 30.06, *p* < 0.001), *KRAS* (42.2% vs. 20.4%, χ^2^ = 25.27, *p* < 0.001), and *BRAF* (9.5% vs. 20.4%, χ^2^ = 17.43, *p* < 0.001), indicating statistically significant differences in mutation prevalence between cohorts. In contrast, mutation frequencies of *NRAS* (4.4% vs. 2.2%, *p* = 0.288) and *ERBB2* (2.5% vs. 1.5%, *p* = 1.000) were not significantly different, as assessed using Fisher’s exact test due to the low expected count. Thus, relative to the clinical cohort, the PDX cohort displayed a significant decrease in *APC*, *TP53*, and *KRAS* mutations, respectively, and a significant increase in *BRAF* mutations. These results are summarized in Table 1.

### 2.2. Comparison of Concordance in Mutational Patterns Between the Clinical Cohort and the PDX Cohort

Subsequently, we comparatively analyzed which mutational combinations are most frequently observed, starting from these six most relevant mutations in CRC. In the clinical cohort, the most prevalent mutation group was the combination of *TP53* and *APC*, accounting for 29.4% of cases. This was followed by the triple *KRAS* and *TP53* and *APC*, with 18.1% of cases, and the co-occurrence of *KRAS* and *APC* mutations, with 12.8% of cases. Notably, in a mere 9.8% of the clinical cohort, no mutations were observed. In contrast, the PDX cohort demonstrated a distinct pattern: no mutations were identified in 29.9% of cases, and while the *TP53* and *APC* group was prevalent at 17.5%, the triple mutation *KRAS* + *TP53* + *APC* and the combination *KRAS* and *APC* were each observed in only 4.4% of cases. Furthermore, single mutations, such as *TP53* and *APC*, were more prevalent in the PDX cohort, with 15.3% and 16.8% prevalence, respectively. Isolated *KRAS* mutations were identified in 6.6% of the PDX cohort, which is slightly higher than the percentage observed in the clinical cohort. In sum, this comparison revealed that while the *TP53* + *APC* combination is the most frequent co-occurrence in both, the PDX cohort has a higher proportion of cases with no mutations and a greater prevalence of single-gene mutations, whereas the public cohort exhibits elevated frequencies of multiple-gene mutation combinations. Chi-square tests were used to compare mutation combination frequencies between clinical and PDX cohorts. Significant differences were observed for *TP53* + *APC* (χ^2^ = 8.61, *p* = 0.003), *KRAS* + *TP53* + *APC* (χ^2^ = 16.41, *p* < 0.001), and *KRAS* + *APC* (χ^2^ = 7.88, *p* = 0.005). A highly significant enrichment of *APC* alone (χ^2^ = 16.17, *p* < 0.001) and of samples with no mutations (χ^2^ = 59.30, *p* < 0.001) was noted in the PDX cohort.

No significant differences were found for *TP53* (χ^2^ = 1.60, *p* = 0.207), *KRAS* + *TP53* (χ^2^ = 0.14, *p* = 0.708), or *KRAS* alone (χ^2^ = 0.41, *p* = 0.521). The results of this analysis are presented in Table 2.

### 2.3. Comparison of the Concordance in Mutational Patterns in the Clinically Significant Mutations (KRAS, BRAF, NRAS, and ERBB2) Between the Clinical Cohort and the PDX Cohort

In the third step of the analysis, the mutational co-occurrence of the following clinically significant mutations in CRC was examined in relation to treatment decisions: *KRAS*, *BRAF*, *NRAS*, and *ERBB2*. In both datasets, cases with no mutations were the most prevalent, comprising 43.7% of the clinical cohort and 55.4% of the PDX cohort. *KRAS* alone is the most frequent mutated gene in both cohorts, occurring in 40.2% of the public database cohort and 20.4% of the PDX cohort. *BRAF* alone is the second most mutated gene, appearing in 8.6% of the clinical cohort and 20.1% of the PDX cohort. The occurrence of mutations in *NRAS* (3.8% in the clinical cohort and 2.2% in the PDX cohort) and *ERBB2* (1.3% in the clinical cohort and 1.4% in the PDX cohort) is relatively infrequent, yet similarly represented. However, while the clinical cohort demonstrates infrequent instances of concomitant mutations (e.g., *KRAS* + *BRAF*, *NRAS* + *ERBB2*), no co-occurrence of these mutations was observed in the PDX cohort. Chi-square analysis showed a significant decrease in the frequency of *KRAS*, *TP53*, and *APC* mutations in the PDX cohort compared to the clinical cohort (*p* < 0.001 for all). In contrast, *BRAF* mutations were significantly enriched in PDXs (*p* < 0.001, χ^2^ = 655.33). The proportion of samples with no detectable mutation was also significantly higher in PDXs (*p* = 0.008, χ^2^ = 7.01). No significant differences were observed for *NRAS* or *ERBB2* mutations (Fisher’s exact test, *p* > 0.05). These results are summarized in Table 3.

We performed pairwise mutual exclusivity analysis using log_2_ odds ratios, *p*-values, and q-values to quantify co-mutation patterns. As expected, *KRAS–BRAF* and *KRAS*–*NRAS* pairs showed strong mutual exclusivity (log_2_ OR < −3, q < 0.001), confirming their roles as alternative effectors in the MAPK pathway. Additional mutually exclusive pairs included *KRAS*–*TP53*, *BRAF*–*ERBB2*, *BRAF*–*NRAS*, and *TP53*–*BRAF* (log_2_ OR < −0.8, q < 0.001). In contrast, significant co-occurrence was observed between *APC*–*TP53*, *APC*–*KRAS*, and *APC*–*NRAS* (log_2_ OR > 0.6, q < 0.001), suggesting cooperative roles in tumorigenesis. No significant association was found for *TP53*–*NRAS*, *KRAS*–*ERBB2*, and *APC*–*ERBB2* (q > 0.05), and they showed no consistent pattern (see Table 4 for details).

## 3. Discussion

We conducted a comparative analysis of the data from the public databases and the data from our collection of PDX models to determine whether PDX models capture the full spectrum of mutations and mutational patterns observed in clinical cases. We also analyzed to which extent PDX models are able to replicate real-world clinical cases.

To address these questions, we first identified the most relevant genes in CRC. These were the protein-altering non-synonymous mutations in *APC* and *TP53*, as well as the clinically relevant mutations for precision therapy affecting *KRAS*, *BRAF*, *NRAS,* and *ERBB2*. In order to ensure consistency and biological relevance in our analysis, we included only mutations classified as “pathogenic” or “likely pathogenic” based on curated clinical significance annotations, double-checked using databases ClinVar and Cosmic. All selected mutations were located within coding regions and had well-documented functional consequences, including nonsense, missense, and frameshift alterations known to affect protein function. The interplay between the genes leading to CRC progression is portrayed in Figure 2.

Following a thorough analysis of the frequency of individual mutations, it was ascertained that *APC*, *TP53*, and *KRAS* were the most prevalent mutations in both groups. This finding is significant because these genes play a pivotal role in the development of CRC. *APC*, a tumor suppressor gene, is involved in the regulation of the Wnt signaling pathway. *TP53* functions as a pivotal tumor suppressor gene, and its mutation plays a critical role in tumor progression [14]. *KRAS* is a proto-oncogene that activates the MAPK signaling pathway, promoting uncontrolled cell proliferation and tumor progression [15]. The similar comparative ranking of these mutations suggests that the PDX model retains key characteristics of a tumor’s genetic landscape. However, the observed discrepancies in mutation frequencies with a decrease in the frequency of *APC*, *TP53*, and *KRAS*, in the PDX cohort compared to the clinical cohort, are indicative of the loss of the heterogeneity of a clinical tumor during modeling [16]. These discrepancies may be attributed to various factors intrinsic to the establishment and maintenance of PDX models. The process of engrafting human tumors into immunocompromised mice can introduce selective pressures that favor the growth of certain subclones over others, potentially leading to an underrepresentation of specific mutations [17]. A striking difference between the two cohorts when comparing the single mutations was that *BRAF* mutations were substantially more prevalent in the PDX cohort, that is, more than double the frequency observed in the clinical cohort. *BRAF* is a well-established oncogene, and its mutation has been identified as a key factor in the development of various cancers. It has been implicated in the activation of downstream EGFR signaling, which promotes cancer cell proliferation and survival [8]. The most plausible explanation for these observed differences is selective engraftment bias, which is the preferential engraftment of more aggressive tumors with stronger growth potential [18]. Through the selection bias during the engraftment process, tumors harboring *BRAF* mutations may have a higher propensity to establish and proliferate in the murine host environment, hence leading to the high number of *BRAF* mutations in the PDX cohort. The high rate of *BRAF*-mutated tumors in PDX models may be attributed to engraftment success, whereby tumors harboring *BRAF* mutations may exhibit heightened proliferative potential, thereby increasing their probability of successful PDX establishment [17,18].

*BRAF* mutation leads to the constitutive activation of the *MAPK/ERK* signaling pathway, independent of upstream growth signals [19]. As a result, tumor cells gain a strong growth and survival advantage, which leads to increased proliferation, enhanced metabolic activity, and resistance to apoptosis [19]. During the PDX transplantation process, only the most adaptable tumor cells survive the engraftment bottleneck, *BRAF*-mut cells often dominate this selection due to their intrinsic growth signaling. After transplantation into immunodeficient mice, tumor cells face stress due to a foreign microenvironment, limited stromal support, and lack of human-specific growth factors, which present some limiting factors to PDX establishment. *BRAF*-mut cells are less dependent on external signals, and are hence able to proliferate autonomously and adapt more readily to the murine host [20].

Moreover, due to the intrinsic characteristics of *BRAF* mutations, *BRAF*-driven tumors in humans often exhibit aggressive phenotypes, which correlate with higher engraftment rates and faster tumor growth in vivo—two essential factors for successful PDX generation [19]. For instance, *BRAF*-mut tumors can upregulate pro-angiogenic factors like *VEGF*, promoting faster vascularization within the graft. Rapid blood vessel formation improves oxygen and nutrient delivery, supporting tumor cell survival and expansion in the new host [20].

A number of studies have identified variations in the engraftment rates of the four consensus molecular subtypes (CMS) of CRC. Specifically, CMS1 and CMS4 have been observed to demonstrate notable advantages [21,22]. This suggests that the PDX model may have certain mutations at higher frequencies than the primary tumors, most likely due to the selective nature of the engraftment process. Similarly demonstrated in the study by Suto et al., Lynch syndrome-associated CRC cases exhibited a higher degree of retention of their histological characteristics during the process of PDX engraftment when compared to sporadic MSI cases [23].

Furthermore, successive passages may result in the loss of subclonal heterogeneity in PDX models. Concurrently, a replacement of human tumor stroma by murine-derived extracellular matrix and stromal cells takes place, which can lead to a reduction in the fidelity to the original tumor. This weakness of PDX models is represented in Figure 3. For instance, a study investigating glioma PDX models demonstrated that while key molecular drivers were retained at frequencies comparable to those in primary tumors, certain genetic alterations were either acquired or lost, indicative of clonal selection processes [24]. The DNA-based copy number profiles indicated that later-passage PDX models exhibited a reduced resemblance to primary tumors [25]. These findings underscore the notion that early-passage PDX models may offer superior fidelity in preserving tumor characteristics [26], a concept that is likely extendable to CRC PDX models. It should be noted that in this study, we used early passage PDX models (passage 0 to 3), since early passage models preserve the original tumor heterogeneity while minimizing clonal drift [27]. Since higher passage numbers may introduce biological divergence due to the loss of most human tumor microenvironment at a higher passage, our consistent use of early-to-mid passages mitigates such concerns and provides a robust basis for the conclusions drawn.

Furthermore, the comparative analysis of co-mutation patterns revealed a remarkable concordance in the relative ranking of mutation co-occurrence patterns between the clinical cohort and the PDX cohort. The frequent co-occurrence of *TP53* and *APC* mutations in both cohorts, as the most common combination, underscores the central role of these mutations in the process of tumorigenesis [28]. The maintenance of this pattern in the PDX cohort further shows that these models accurately represent the core genetic alterations that are essential for tumor growth and progression.

Nevertheless, a notable difference between the two cohorts is that the PDX cohort exhibited an overrepresentation of single mutations and a striking underrepresentation of complex mutation patterns. This phenomenon can be explained by the selective advantage that these mutations provide during the process of engraftment. It is conceivable that these mutations, in and of themselves, are sufficient to allow for growth and survival when xenografted.

It has been observed that the primary tumor exhibits a polyclonal pattern of mutations, while models, even in early passages, demonstrate an oligoclonal pattern at best. Consequently, minor subclones are identified in whole tumor analyses, yet their presence is absent in pure model settings [29].

This has also been shown in several publications; in one study, the authors observed that a significant proportion of heterogeneous primary tumors gave rise to monoclonal PDXs (60% or more), while the remainder gave rise to polyclonal PDXs, owing to the presence of distinct subclones within the regions of origin in the patient tumors. This finding suggests the possibility of a bottleneck event during PDX engraftment [25]. Hynds et al. further demonstrate that selective pressure during initial PDX engraftment and genomic evolution during the passaging through mice determine the genomic characteristics of the PDXs. However, the accumulation of mutations during expansion in mice contributed less to the overall genomic distance of PDX models from primary tumors compared to the initial bottlenecking events. This finding suggests that PDXs exhibit a lower degree of heterogeneity compared to original patient tumors [30].

A notable distinction between the two cohorts pertains to the higher prevalence of “no mutation” (30.2%) in the PDX cohort. This phenomenon may be attributable to several factors, including the potential loss of certain mutations during the engraftment process, the possibility of selection bias favoring clones that are more suitable for engraftment, and limitations in sequencing depth or scope during PDX model characterization [29].

It is also imperative to acknowledge that the clinical cohort from public databases encompasses a more expansive and heterogeneous patient population, whereas the PDX models derive from a subset of tumors that successfully engrafted, which may introduce additional selection biases. The limited sample size of the PDX cohort (*n* = 139) may contribute to these observed differences in comparison to the clinical cohort (*n* = 7984), as smaller cohorts are more susceptible to sampling and statistical variability, which may affect the reliability of frequency comparisons.

Finally, we conclude that no relevant mutation or mutational groups are entirely absent from the PDX cohort in comparison to the clinical cohort. The PDX models appear to exhibit a comprehensive retention of the pertinent mutations and mutation group combinations identified within the clinical cohort. However, they may underrepresent complex mutational interactions while favoring simpler genetic profiles, suggesting a tendency to lose intricate mutation patterns while preserving key mutations of the original tumor. It is important to note some potential limitations to our study. First, since the study focuses on a limited set of six key mutations, we might potentially be overlooking other significant genetic alterations that contribute to CRC heterogeneity. Second, potential differences in sample collection methods, data curation, and quality control between the diverse public database sources of the clinical sample could confound the comparative analysis. Furthermore, possible methodological differences inherent to data aggregation from public databases versus PDX data might complicate direct comparisons. Third, the disparity in sample sizes between the extensive clinical cohort and the smaller, institution-specific PDX cohort may introduce statistical variability and bias, potentially skewing the comparative analyses.

In order to overcome the limitations currently observed in CRC modeling, future research could enhance early-passage PDX models by optimizing engraftment protocols and minimizing selective pressures that favor dominant clones [12]. In addition, integrating advanced genomic and multi-omic techniques to monitor clonal evolution over time would provide insights into the dynamics of subclonal populations, enabling targeted interventions to maintain the original heterogeneity. Standardizing tissue sampling and sequencing protocols across both primary and xenograft specimens would further ensure consistent and comprehensive profiling, ultimately improving the translational relevance and predictive accuracy of PDX models in personalized cancer therapy [12].

## 4. Materials and Methods

### 4.1. PDX Cohort Establishment

Over the course of the past two decades, a consecutive series of biomaterials obtained from patients who underwent surgical procedures at the University Medical Center Rostock have contributed to this extensive biobanking and tumor-modeling initiative. Samples were collected in strict accordance with standard operating procedures, and the HROC collection represents one of the most extensive single-center model assortments from consecutive CRC cases worldwide. The generation of PDX models from all histological sides, stages, and known subtypes of CRC is possible. The PDX models were established from freshly resected solid tumor tissues, which were mechanically fragmented and subcutaneously implanted into either NSG (NOD-scid IL2Rγ^null^) or NMRI nude mice. This section provides a thorough and comprehensive overview of the subject matter. All models were propagated via subcutaneous engraftment and cryopreserved after expansion in the BioBank Rostock as part of the HROC collection for future use [31,32].

From a molecular perspective, the models encompass chromosomal instability (CIN), CpG island methylator phenotype (CIMP), microsatellite instability-low (MSI-L), and microsatellite instability-high (MSI-H) cases, both sporadic and those associated with Lynch syndrome (LS). Additionally, the models represent a broad spectrum of mutations in key colorectal cancer (CRC) driver genes, including *KRAS*, *BRAF*, *TP53*, *PIK3CA*, and *APC*.

To minimize genomic drift and preserve tumor fidelity, all experiments were conducted using early-passage (P0–P3) PDX models in passage-controlled cohorts, despite not analyzing passage number as an independent variable due to limited sample size. As higher passage numbers can introduce biological divergence—particularly due to the loss of most human tumor microenvironment—we consistently used early passage models to mitigate this concern. A detailed list of the HROC models used in this study, along with their characteristics, is provided in Appendix A.

### 4.2. Clinical Cohort Establishment

To establish our clinical cohort, we integrated data from 20 publicly available colorectal cancer datasets. These encompassed a diverse range of tumor types, stages, and study contexts, including early neoplastic lesions (HTAN Vanderbilt), metastatic disease (MSK, Cancer Cell 2018), and organoid-based drug response models (MSK, Nat Med 2019). Population-specific cohorts were also included, such as studies investigating racial disparities in metastatic colorectal cancer (MSK, 2020). The cohort draws on major genomic consortia (e.g., TCGA and CPTAC) as well as institutionally led studies from MSK, Genentech, and Sidra-LUMC, among others. This comprehensive approach ensured both biological heterogeneity and technical diversity, supporting robust analyses of somatic mutations and clinical associations. The complete list of datasets is detailed in Table 5. All cases with missing mutation information for any of the six genes analyzed in our study were excluded.

To ensure robust data processing, we established a key criterion: the detection frequency of candidate high-frequency mutated genes must reach at least 80% of the total sample size. This threshold minimizes potential batch effects and enhances the reliability of the analysis. cBioPortal (https://www.cbioportal.org/) is a web-based bioinformatics platform for integrating and visualizing genomic data from cancer studies, facilitating the exploration of gene mutations, copy number alterations, and clinical data. We used cBioPortal to visualize highly mutated genes and GenVisR (version 4.0.3) to analyze mutation data, clinical phenotypes, and pathological features across 15 datasets, systematically examining high-frequency gene mutation patterns. This systematic approach enabled the construction of a unified and analytically rigorous clinical cohort for investigating high-frequency gene mutation landscapes across CRC.

### 4.3. Mutation Analysis of the PDX Cohort

Whole exome sequencing (WES) and targeted panel sequencing were conducted [32]. In short, genomic DNA was extracted from cultured CRC cell lines using the Precellys Tissue DNA Kit (Bertin, Rockville, MD, USA), following the manufacturer’s instructions. The DNA had to meet a minimum concentration of 25 ng/μL and a DNA Integrity Number of at least 5 to ensure reliable sequencing. Next-generation sequencing was performed by Centogene AG using the Illumina NextSeq500 platform (Illumina, San Diego, CA, USA). HiSeq4000 and NovaSeq instruments using the Twist Human Core Exome Plus enrichment kit (Twist Bioscience, San Francisco, CA, USA), which targets approximately 36.5 Mb of the human coding exome were used for WES. A clinically validated solid tumor gene panel was used for targeted panel sequencing, covering all coding regions of 105 known cancer-related genes and hotspot regions in another 146 oncogenes and tumor suppressor genes. Library preparation was performed with the Twist Bioscience enzymatic fragmentation and hybrid capture kit. Sequencing generated 2 × 150 bp paired-end reads, which were demultiplexed using Illumina’s bcl2fastq software (v2.17.1.14). Reads were aligned to the human reference genome (GRCh37/hg19) using either Bowtie2 bwa-mem. Aligned reads were sorted (samtools v1.11), de-duplicated (PicardTools v2.23.8), and recalibrated prior to variant calling. Germline variants were identified using Sentieon HaplotypeCaller, and somatic variants were detected with either the TNhaplotyper or the Strelka Somatic Pipeline (v2.9.2). Stringent quality control measures were applied to all samples, including Phred-scaled read quality scores >30, coverage of ≥20× across target regions, and assessment of on- and off-target alignments. Variants were filtered to include only protein-coding mutations with “PASS” status, allele frequency > 5%, quality score > 50, and at least 20 supporting reads in tumor samples. Final variant calls were normalized, annotated using snpEff, and stored in a dedicated genomic database.

A 10% cutoff for mutation frequency was determined empirically based on the distribution of mutation frequencies observed in our dataset. Notably, the top three most frequently mutated genes in the public database all exhibited mutation frequencies above 10%, while the frequency dropped sharply below 10% for the fourth-ranked gene. Therefore, this threshold effectively distinguishes a small group of genes with markedly higher mutation rates, which we classified as highly relevant.

### 4.4. Statistical Analysis

All statistical analyses were performed using SPSS Statistics (15.0; IBM, Armonk, NY, USA). To compare mutation or pattern frequencies between the clinical cohort and the PDX cohort, we used the Chi-square (χ^2^) test for categorical variables where the expected cell count was ≥ 5. For comparisons involving rare mutations or where the expected frequency in any cell was < 5, we applied the Fisher’s exact test. For co-recurrence and mutual exclusivity analysis of gene mutations within the clinical cohort, we constructed 2 × 2 contingency tables for selected gene pairs. The presence or absence of each gene mutation was cross-tabulated, and Fisher’s exact test (one-sided) was used to assess whether the observed co-occurrence was significantly less than expected under the assumption of independence. A *p*-value less than 0.05 was considered statistically significant.

## 5. Conclusions

In summary, PDX models effectively capture the mutational landscape and key genetic interactions of the primary tumor and are able to reflect the fundamental drivers of tumorigenesis. However, a few discrepancies were observed, including the lower frequency of complex multi-mutational patterns and the higher proportion of cases with no detected mutations. These observations hint towards potential limitations, including clonal selection during transplantation and the predominance of a polyclonal mutation pattern throughout the tumor, as opposed to the oligoclonal mutation pattern observed in models. Despite these variations, the trends observed within the PDX cohort demonstrate its ability to reflect the genetic complexity of primary tumors, establishing it as a valuable resource for the study of oncogenic pathways and therapeutic responses. To enhance the translational relevance of PDX models, it is crucial to mitigate the biases associated with transplantation and to expand these models with additional systems to comprehensively address tumor heterogeneity.

## Figures and Tables

**Figure 1 ijms-26-05111-f001:**
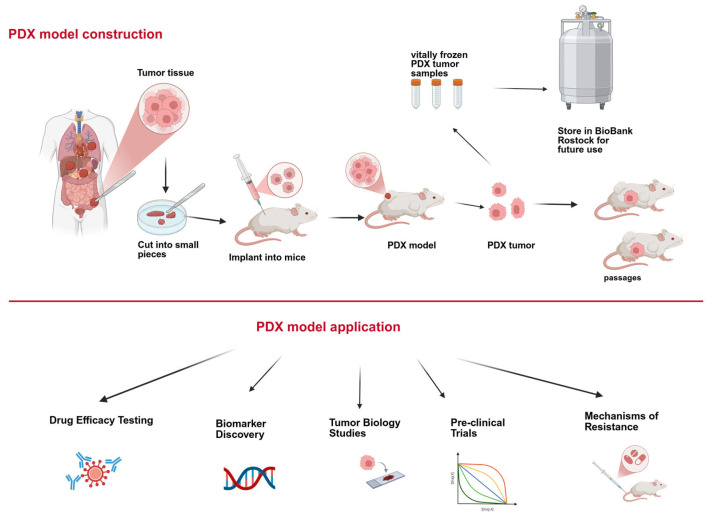
Construction and application of the PDX models. PDX models were constructed by transplanting patient-derived tumor tissue into immunodeficient mice. They offer the advantage of delivering expandable tumor tissue for a variety of subsequent research applications.

**Figure 2 ijms-26-05111-f002:**
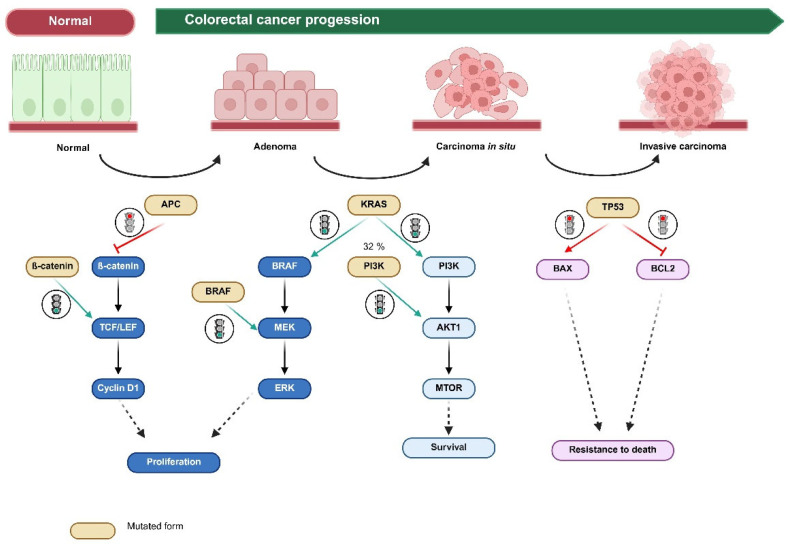
Interplay between genes in CRC. This figure offers a schematic overview of CRC progression from normal colonic epithelium through adenoma to invasive carcinoma, illustrating morphological changes and key gene mutations.

**Figure 3 ijms-26-05111-f003:**
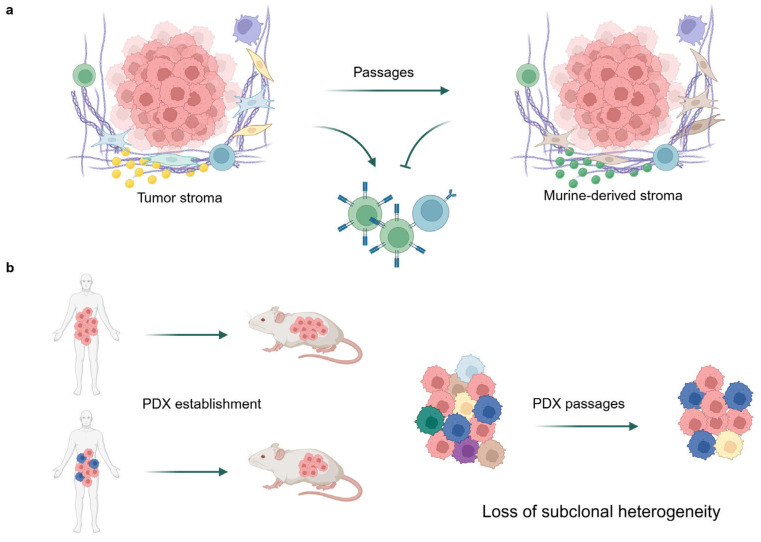
The weaknesses existing in current PDX models. (**a**) Tumor stroma tends to be replaced by murine-derived ECM and stromal cells after several passages, which hinders immune cell activation without human cytokine secretion. (**b**) Loss of subclone heterogeneity during the establishment and passage of PDX.

**Table 1 ijms-26-05111-t001:** Mutation frequency comparison: clinical cohort vs. PDX cohort.

Mutation In	Clinical Cohort		PDX Cohort		χ^2^	*p*	Test Used
	*n* = 7936	(%)	*n* = 137	(%)			
*APC*	5366	67.6	59	43.1	35.72	<0.001	Chi-square
*TP53*	5179	65.3	5179	42.3	30.06	<0.001	Chi-square
*KRAS*	3347	42.2	28	20.4	25.27	<0.001	Chi-square
*BRAF*	750	9.5	28	20.4	17.43	<0.001	Chi-square
*NRAS*	346	4.4	3	2.2	NA	0.288	Fisher’s exact
*ERBB2*	195	2.5	2	1.5	NA	1.000	Fisher’s exact

**Table 2 ijms-26-05111-t002:** Differences in mutation patterns between the clinical cohort and the PDX cohort.

Mutation In	Clinical Cohort		PDX Cohort		χ^2^	*p*	Test Used
	*n* = 7936	(%)	*n* = 137	(%)			
*TP53* + *APC*	2332	29.4	24	17.5	8.61	0.003	Chi-square
*KRAS* + *TP53* + *APC*	1439	18.1	6	4.4	16.41	<0.001	Chi-square
*KRAS* + *APC*	1015	12.8	6	4.4	7.88	0.005	Chi-square
*TP53*	911	11.5	21	15.3	1.60	0.207	Chi-square
No mutation	766	9.7	41	29.9	59.30	<0.001	Chi-square
*APC*	580	7.3	23	16.8	16.17	<0.001	Chi-square
*KRAS* + *TP53*	497	6.3	7	5.1	0.14	0.708	Chi-square
*KRAS*	396	5.0	9	6.6	0.41	0.521	Chi-square

**Table 3 ijms-26-05111-t003:** Differences in mutation patterns in the clinically significant genes between the clinical cohort and the PDX cohort.

Mutation In	Clinical Cohort		PDX Cohort		χ^2^	*p*	Test Used
	*n* = 7936	%	*n* = 137	%			
No mutation	3474	43.7	76	55.4	7.01	0.008	Chi-Square
*KRAS*	3192	40.2	28	20.4	3142.43	<0.001	Chi-Square
*BRAF*	689	8.6	28	20.4	655.33	<0.001	Chi-Square
*NRAS*	303	3.8	3	2.19	NA	0.450	Fisher’s Exact
*ERBB2*	104	1.3	2	1.46	NA	0.701	Fisher’s Exact
*ERBB2 + KRAS*	83	1.05	0		NA	0.406	Fisher’s Exact
*KRAS + BRAF*	45	<1	0		NA	1.000	Fisher’s Exact
*NRAS + KRAS*	26	<1	0		NA	1.000	Fisher’s Exact
*NRAS + BRAF*	12	<1	0		NA	1.000	Fisher’s Exact
*NRAS + ERBB2*	4	<1	0		NA	1.000	Fisher’s Exact
*BRAF + ERBB2*	2	<1	0		NA	1.000	Fisher’s Exact
*NRAS + BRAF + ERBB2*	1	<1	0		NA	1.000	Fisher’s Exact
*KRAS + BRAF + ERBB2*	1	<1	0		NA	1.000	Fisher’s Exact

**Table 4 ijms-26-05111-t004:** Mutual exclusivity analysis between pairs of genes, using 2 × 2 contingency tables and associated *p*-values.

A	B	Neither	A Not B	B Not A	Both	Log^2^ Odds Ratio	*p*-Value	q-Value	Tendency
*KRAS*	*BRAF*	3885	3301	704	46	<−3	<0.001	<0.001	Mutual exclusivity
*APC*	*BRAF*	2074	5112	496	254	−2.267	<0.001	<0.001	Mutual exclusivity
*KRAS*	*NRAS*	4269	3321	320	26	<−3	<0.001	<0.001	Mutual exclusivity
*APC*	*TP53*	1162	1595	1408	3771	0.964	<0.001	<0.001	Co-occurrence
*TP53*	*KRAS*	1346	3243	1411	1936	−0.812	<0.001	<0.001	Mutual exclusivity
*APC*	*KRAS*	1677	2912	893	2454	0.662	<0.001	<0.001	Co-occurrence
*TP53*	*BRAF*	2386	4800	371	379	−0.978	<0.001	<0.001	Mutual exclusivity
*APC*	*NRAS*	2510	5080	60	286	1.236	<0.001	<0.001	Co-occurrence
*BRAF*	*ERBB2*	6995	746	191	4	−2.348	<0.001	<0.001	Mutual exclusivity
*BRAF*	*NRAS*	6853	737	333	13	−1.462	<0.001	<0.001	Mutual exclusivity
*TP53*	*ERBB2*	2669	5072	88	107	−0.644	0.003	0.004	Mutual exclusivity
*NRAS*	*ERBB2*	7400	341	190	5	−0.808	0.284	0.356	
*TP53*	*NRAS*	2642	4948	115	231	0.101	0.564	0.63	
*APC*	*ERBB2*	2503	5238	67	128	−0.131	0.588	0.63	
*KRAS*	*ERBB2*	4478	3263	111	84	0.055	0.826	0.826	

**Table 5 ijms-26-05111-t005:** List of public database used to establish the clinical cohort.

Project Name	Sample Count	Source	Year	Reference Genome	Cancer Type
Colorectal Adenocarcinoma (DFCI/Orion)	74	DFCI/Orion	2024	GRCh37/hg19	CRC
Colorectal Adenocarcinoma (MSK, Nat Commun)	179	MSK	2022	GRCh37/hg19	CRC
Colorectal Cancer (MSK, JNCI)	1516	MSK	2021	GRCh37/hg19	CRC
Colorectal Adenocarcinoma (DFCI, Cell Reports)	619	DFCI	2016	GRCh37/hg19	CRC
Colorectal Adenocarcinoma (Genentech, Nature)	74	Genentech	2012	GRCh37/hg19	CRC
Colorectal Adenocarcinoma (TCGA, Firehose Legacy)	640	TCGA		GRCh37/hg19	CRC
Colorectal Adenocarcinoma (TCGA, Nature)	276	TCGA	2012	GRCh37/hg19	CRC
Colorectal Adenocarcinoma (TCGA, PanCancer Atlas)	594	TCGA		GRCh37/hg19	CRC
Colorectal Adenocarcinoma Triplets (MSK, Genome Biol)	138	MSK	2014	GRCh37/hg19	CRC
Colorectal Cancer (CAS Shanghai, Cancer Cell)	146	CAS Shanghai	2020	GRCh37/hg19	CRC
Colorectal Cancer (MSK, Cancer Discovery)	22	MSK	2022	GRCh37/hg19	CRC
Colorectal Cancer (MSK, Gastroenterology)	471	MSK	2020	GRCh37/hg19	CRC
Colorectal Cancer (MSK, JCO Precis Oncol)	47	MSK	2022	GRCh37/hg19	CRC
Disparities in metastatic colorectal cancer (MSK)	64	MSK	2020	GRCh37/hg19	CRC
Metastatic Colorectal Cancer (MSK, Cancer Cell)	1134	MSK	2018	GRCh37/hg19	CRC
Pre-cancer Colorectal Polyps (HTAN Vanderbilt, Cell)	61	HTAN Vanderbilt	2021	GRCh37/hg19	CRC
Colon Adenocarcinoma (CPTAC, GDC)	109	CPTAC		GRCh38/hg38	Colon Cancer
Colon Adenocarcinoma (CaseCCC, PNAS)	29	CaseCCC	2015	GRCh37/hg19	Colon Cancer
Colon Adenocarcinoma (TCGA, GDC)	463	TCGA		GRCh38/hg38	Colon Cancer
Colon Cancer (CPTAC-2 Prospective, Cell)	110	CPTAC-2	2019	GRCh37/hg19	Colon Cancer
Colon Cancer (Sidra-LUMC AC-ICAM, Nat Med)	348	Sidra-LUMC	2023	GRCh37/hg19	Colon Cancer
Rectal Cancer (MSK, Nature Medicine)	788	MSK	2022	GRCh37/hg19	Rectal Cancer
Rectal Cancer (MSK, Nature Medicine)	339	MSK	2019	GRCh37/hg19	Rectal Cancer
Colorectal Cancer Radiation (MSK)	48	MSK	2024	GRCh37/hg19	Rectal Cancer
Rectal Adenocarcinoma (TCGA, GDC)	171	TCGA		GRCh38/hg38	Rectal Cancer
Appendiceal Cancer (MSK, J Clin Oncol)	273	MSK	2022	GRCh37/hg19	Appendiceal Cancer

## Data Availability

The data and materials are available from the corresponding author upon reasonable request.

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
