# Peer review of "Mutational Patterns in Colorectal Cancer: Do PDX Models Retain the Heterogeneity of the Original Tumor?"

_ijms, 2025, doi:10.3390/ijms26115111_

Round 1

Reviewer 1 Report

Comments and Suggestions for Authors

The manuscript addresses a critical issue in colorectal cancer (CRC) research methodology by examining the genetic profiling of patient-derived xenografts (PDX) and assessing how accurately these models reflect patient tumors at the population level. By comparing the mutational profiles of the PDX models with publicly available datasets, the study provides valuable insights into the reliability of these models for CRC research. A detailed analysis of both individual and co-occurring mutations in CRC has allowed the authors to identify key patterns and discrepancies between datasets, offering a deeper understanding of CRC's genetic landscape and highlighting potential challenges and considerations for using PDX models in personalized medicine.

While the study design is solid and the paper is generally well-written, there are a couple of important issues that need to be addressed before publication:

My primary concern lies in the lack of detailed data and explanation about the types of mutations included in the analysis, which impacts the overall conclusions of the study. Simply reporting the number of alterations in a gene is not sufficient unless it is clear that the mutations are categorized consistently in terms of their pathogenicity and the functional relevance of their consequences.

The section on CRC cohorts needs to be expanded to include essential clinical characteristics, histochemical, and molecular data of the samples. While these data have been previously published and referenced, providing them in the manuscript is crucial for the study's context and interpretation.

There are several minor issues that the authors should address to improve the manuscript's presentation:

Recheck the numbers in sections 2.1 and 2.2, as well as in tables 1 and 2, as there are discrepancies.

Figure 1 lacks clarity and informativeness. It should either be improved or removed.

The term "HROC" should be defined in section 2.1.

Consider clarifying the subtitles in sections 2.2 and 2.3. Additionally, all subtitles should refer to mutations, not genes.

Table 3 is missing from the document.

In Figure 2, the relationship between tumor stage shown at the top and the mutations in the diagram below should be clarified.

Author Response

Response to Reviewers

We would like to thank all reviewers for their thorough evaluation of our manuscript and their insightful comments. We have addressed each point carefully and made revisions to improve the quality, clarity, and scientific rigor of the manuscript. Below, we provide a detailed point-by-point response to each reviewer’s suggestions.

Reviewer 1

We are grateful for the reviewer’s positive evaluation of our study and valuable suggestions. Below are our specific responses:

The manuscript addresses a critical issue in colorectal cancer (CRC) research methodology by examining the genetic profiling of patient-derived xenografts (PDX) and assessing how accurately these models reflect patient tumors at the population level. By comparing the mutational profiles of the PDX models with publicly available datasets, the study provides valuable insights into the reliability of these models for CRC research. A detailed analysis of both individual and co-occurring mutations in CRC has allowed the authors to identify key patterns and discrepancies between datasets, offering a deeper understanding of CRC's genetic landscape and highlighting potential challenges and considerations for using PDX models in personalized medicine. While the study design is solid and the paper is generally well-written, there are a couple of important issues that need to be addressed before publication:

  1. My primary concern lies in the lack of detailed data and explanation about the types of mutations included in the analysis, which impacts the overall conclusions of the study. Simply reporting the number of alterations in a gene is not sufficient unless it is clear that the mutations are categorized consistently in terms of their pathogenicity and the functional relevance of their consequences.

Response:  We appreciate this important point. We have now expanded the Methods section to include a detailed description of the mutation types analyzed  and explained how these were categorized based on their pathogenicity and functional relevance. Databases such as ClinVar and COSMIC were used to support this classification.

  1. The section on CRC cohorts needs to be expanded to include essential clinical characteristics, histochemical, and molecular data of the samples. While these data have been previously published and referenced, providing them in the manuscript is crucial for the study's context and interpretation.

Response: We have now added a concise summary of the key clinical, histochemical, and molecular characteristics of the CRC samples , which we suggest to be included as a Supplementary Table (i.e. Supplementary Table 1). This data completes thus the mutational data from the models, which we added after careful revision as Supplementary Table 2 into the revised and improved version of our manuscript.

There are several minor issues that the authors should address to improve the manuscript's presentation:

  1. Recheck the numbers in sections 2.1 and 2.2, as well as in tables 1 and 2, as there are discrepancies.

Response: Since there were novel data in the public databases, the analysis on the clinical cohort was updated and re-done. We reviewed all the novel numerical data and eliminated any inconsistencies found in the previous version’s Tables. The revised Tables now reflect up-to-date, accurate and consistent information.

  1. Figure 1 lacks clarity and informativeness. It should either be improved or removed.

Response: We have revised Figure 1 to improve visual clarity, enhance labeling, and add a more descriptive figure legend. The updated version is now included in the revised manuscript.

  1. The term "HROC" should be defined in section 2.1.

Response: We have added the full name—Hansestadt Rostock CRC (HROC)—at its first mention in Section 2.1.

  1. Consider clarifying the subtitles in sections 2.2 and 2.3. Additionally, all subtitles should refer to mutations, not genes.

Response: We have revised the subtitles throughout these sections to accurately reflect that the analysis pertains to mutations, not genes.

  1. Table 3 is missing from the document.

Response: Thank you for pointing this out; this mistake has been solved in the revised manuscript.

  1. In Figure 2, the relationship between tumor stage shown at the top and the mutations in the diagram below should be clarified.

Response: Thank you for this valuable hint. We have now included Table 3 in the revised manuscript

Reviewer 2 Report

Comments and Suggestions for Authors

Dear Authors,

First of all, congratulations for your interesting work. I hope that my hints will help you in the next steps of improvement and the final manuscript will be really valuable for the readers.

It might be a good idea to explain the limitations of the study, even as a bullet-points, as well as potential future directions of the research. 

Moreover, gene names should be written in italics, according to the rules of genetics. Please, familiarise yourself with the rules and change the manuscript accordingly. Examples of rules summary can be found on websites such as: https://www.gmb.org.br/geneprotein-nomenclature-guidelines or https://academic.oup.com/molehr/pages/Gene_And_Protein_Nomenclature 

Finally, I would like to thank you for the excellent figures and graphs you have prepared for the document, they enhance the value of your work and facilitate the understanding process.

Author Response

Response to Reviewers

We would like to thank all reviewers for their thorough evaluation of our manuscript and their insightful comments. We have addressed each point carefully and made revisions to improve the quality, clarity, and scientific rigor of the manuscript. Below, we provide a detailed point-by-point response to each reviewer’s suggestions.

Reviewer 2

Dear Authors,

First of all, congratulations for your interesting work. I hope that my hints will help you in the next steps of improvement and the final manuscript will be really valuable for the readers.

  1. It might be a good idea to explain the limitations of the study, even as a bullet-points, as well as potential future directions of the research. 

Response: We have added a new part at the end of the Discussion, which outlines        the main limitations of our approach such as small PDX sample size and potential      selection bias. We also provided suggestions for future studies, including the            integration of additional systems.

  1. Moreover, gene names should be written in italics, according to the rules of genetics. Please, familiarise yourself with the rules and change the manuscript accordingly. Examples of rules summary can be found on websites such as: https://www.gmb.org.br/geneprotein-nomenclature-guidelines or https://academic.oup.com/molehr/pages/Gene_And_Protein_Nomenclature 

Response: We have carefully revised the entire manuscript and updated all gene names to conform to accepted genetic nomenclature, using italics where appropriate.

  1. Finally, I would like to thank you for the excellent figures and graphs you have prepared for the document, they enhance the value of your work and facilitate the understanding process.

Response: We sincerely thank the reviewer for this positive feedback. We are pleased to hear the visual elements were helpful in communicating our findings.

Reviewer 3 Report

Comments and Suggestions for Authors

The paper "Mutational Patterns in Colorectal Cancer: Do PDX models retain the heterogeneity of the original tumor?" compares mutation profiles between patient-derived xenograft models and clinical samples from public databases. The study's focus on evaluating PDX models as representatives of CRC genetic landscapes is potentially valuable for cancer research. However, the paper lacks critical methodological details about PDX model construction and NGS sequencing parameters. Statistical analysis is insufficient, with no confidence intervals or significance testing for the observed mutation frequency differences. The discussion oversimplifies the mechanism behind BRAF mutation enrichment in PDX models and doesn't address the impact of PDX passage number on results. Additionally, there are inconsistencies in reported mutation frequencies between different sections. Addressing these issues by providing more comprehensive methods, adding statistical rigor, and deepening the mechanistic discussion is necessary before this comparative study can make a meaningful contribution to the field.

"Among these, patient-derived xenografts (PDXs) involve transplanting tumor tissues into immunocompromised mice, preserving tumor heterogeneity and microenvironmental interactions."
The authors only mention the advantages of PDX models without mentioning their limitations, such as differences between immunodeficient mouse environments and human tumor microenvironments. The introduction should briefly mention these limitations to provide context for later interpretation of results.

"Our analysis addresses two main questions: first, whether PDX models accurately represent the complexity of CRC seen in patients, and second, whether they capture all the important subgroups of CRC."
These questions are too broad and vague.

Introduction last paragraph: repeats information about PDX model construction already mentioned, adding no new information.
"The PDX models from the BioBank...": Despite citing previous literature, authors should briefly describe key steps in PDX model construction, including mouse strain used, implantation method (subcutaneous or orthotopic), and passage number, as these factors may affect mutation retention.

"We statistically analyzed the NGS...": Authors fail to provide key NGS sequencing parameters such as sequencing platform, depth, coverage area (whole exome or targeted gene panel), variant detection algorithms and filtering criteria.
Statistical analysis methods are insufficiently described:
Beyond mentioning UpSet plots for visualization, I didn’t see specify statistical tests used to assess significance of mutation frequency differences between cohorts.

"with genes exhibiting a mutation frequency of at least 10% classified as highly relevant."
Authors don't explain why 10% was chosen as threshold, or whether this choice was based on biological significance or statistical considerations.

"Thus, relative to the clinical cohort, the PDX cohort displayed a decrease of 23.4, 20.7, and 20.8 percentage points in APC, TP53, and KRAS mutations, respectively, and an 11.0 percentage point increase in BRAF mutations."
Confidence intervals should be provided for these differences, especially given the small PDX cohort size which may have substantial sampling error.

"However, while the clinical cohort demonstrates infrequent instances of concomitant mutations (e.g., KRAS + BRAF, NRAS + ERBB2), no co-occurrence of these mutations was observed in the PDX cohort."
For known mutually exclusive mutation pairs (e.g., KRAS and BRAF), specific mutual exclusivity analysis should be performed, with calculation of exclusivity coefficients, rather than just descriptive reporting.
KRAS mutation frequencies reported in section 2.3 (40.2% and 20.1%) differ slightly from those in section 2.1 (40.9% and 20.1%) without explanation.

"The most plausible explanation for these observed differences is selective engraftment bias, which is the preferential engraftment of more aggressive tumors with stronger growth potential [16]."
Discussion attributes BRAF mutation enrichment simply to "selective engraftment bias" without exploring how BRAF mutations specifically affect cellular behavior to promote PDX establishment.

"Furthermore, successive passages may result in the loss of subclonal heterogeneity in PDX models."
The paper mentions passage issues multiple times but doesn't specify which passage of PDX models was used in this study, nor discuss how passage number specifically affected these results, making this discussion theoretical rather than practical.
"Among these, PDXs involve transplanting tumor tissues into immunocompromised mice, preserving tumor heterogeneity and microenvironmental interactions."

Author Response

Response to Reviewers

We would like to thank all reviewers for their thorough evaluation of our manuscript and their insightful comments. We have addressed each point carefully and made revisions to improve the quality, clarity, and scientific rigor of the manuscript. Below, we provide a detailed point-by-point response to each reviewer’s suggestions.

Reviewer 3:

            Comments and Suggestions for Authors

The paper "Mutational Patterns in Colorectal Cancer: Do PDX models retain the heterogeneity of the original tumor?" compares mutation profiles between patient-derived xenograft models and clinical samples from public databases. The study's focus on evaluating PDX models as representatives of CRC genetic landscapes is potentially valuable for cancer research. However, the paper lacks critical methodological details about PDX model construction and NGS sequencing parameters. Statistical analysis is insufficient, with no confidence intervals or significance testing for the observed mutation frequency differences. The discussion oversimplifies the mechanism behind BRAF mutation enrichment in PDX models and doesn't address the impact of PDX passage number on results. Additionally, there are inconsistencies in reported mutation frequencies between different sections. Addressing these issues by providing more comprehensive methods, adding statistical rigor, and deepening the mechanistic discussion is necessary before this comparative study can make a meaningful contribution to the field.

  1. "Among these, patient-derived xenografts (PDXs) involve transplanting tumor tissues into immunocompromised mice, preserving tumor heterogeneity and microenvironmental interactions."  The authors only mention the advantages of PDX models without mentioning their limitations, such as differences between immunodeficient mouse environments and human tumor microenvironments. The introduction should briefly mention these limitations to provide context for later interpretation of results.

Response: We agree and have revised the Introduction to include a brief discussion of the known limitations of PDX models, including the absence of a functional immune system and differences in stromal components.

  1. "Our analysis addresses two main questions: first, whether PDX models accurately represent the complexity of CRC seen in patients, and second, whether they capture all the important subgroups of CRC." These questions are too broad and vague.

Response: We have refined the research questions in the Introduction to be more specific. It now reads: “While PDX models are valuable tools in the field of cancer research, it is important to acknowledge their limitations. A significant drawback of this approach is the progressive replacement of the human tumor stroma by murine-derived stromal cells. This alteration of the tumor microenvironment may have a substantial impact on therapeutic responses [12]. Furthermore, disparities in drug metabolism and molecular signaling pathways between mice and humans can limit the translational relevance of findings derived from PDX models [12]. The vasculature that develops within PDX tumors is of murine origin, potentially influencing tumor perfusion and the efficacy of drug delivery. Furthermore, the interactions between human tumor cells and surrounding murine tissues do not fully recapitulate the complexity of human tumor–host dynamics [13]. Finally, the variability in mouse strains and experimental protocols across different research centers contributes to limited reproducibility and consistency in experimental outcomes [13].”

  1. Introduction last paragraph: repeats information about PDX model construction already mentioned, adding no new information.

Response: We removed the redundant information in the final paragraph of the Introduction as requested.

  1. "The PDX models from the BioBank...": Despite citing previous literature, authors should briefly describe key steps in PDX model construction, including mouse strain used, implantation method (subcutaneous or orthotopic), and passage number, as these factors may affect mutation retention.

Response: We have expanded the Methods section to include these essential details: NSG and NMRI-nude mice were used; tumors were implanted subcutaneously; and all PDX mutation data used in this study were generated from passages up to T3, most in T0-T2.

  1. "We statistically analyzed the NGS...": Authors fail to provide key NGS sequencing parameters such as sequencing platform, depth, coverage area (whole exome or targeted gene panel), variant detection algorithms and filtering criteria.

Response: We thank the reviewer for this important comment. A detailed description of the NGS strategy has been added to the materials and methods section as requested.

  1. Statistical analysis methods are insufficiently described: Beyond mentioning UpSet Suplots for visualization, I didn’t see specify statistical tests used to assess significance of mutation frequency differences between cohorts.

Response: Thank you very much for your valuable suggestion. We have carefully revised the manuscript to address this issue. Specifically, we have now provided a detailed description of the statistical analysis methods used to assess differences in mutation frequencies between cohorts. In particular, we applied the Chi-square test and Fisher’s exact test, as appropriate, to evaluate statistical significance. These methods are now clearly described in the Statistical Analysis section, and the corresponding results have been updated accordingly in the Results section.

  1. "with genes exhibiting a mutation frequency of at least 10% classified as highly relevant." Authors don't explain why 10% was chosen as threshold, or whether this choice was based on biological significance or statistical considerations.

Response: Thank you for your insightful comment. We agree that clarification was needed regarding the rationale for selecting the 10% mutation frequency threshold. We added the following to the Materials and Methods section: “The 10% cutoff was determined empirically based on the distribution of mutation frequencies observed in our dataset. Notably, the top three most frequently mutated genes in the public database all exhibited mutation frequencies above 10%, while the frequency dropped sharply below 10% for the fourth-ranked gene. Therefore, this threshold effectively distinguishes a small group of genes with markedly higher mutation rates, which we classified as highly relevant.”

  1. "Thus, relative to the clinical cohort, the PDX cohort displayed a decrease of 23.4, 20.7, and 20.8 percentage points in APC, TP53, and KRAS mutations, respectively, and an 11.0 percentage point increase in BRAF mutations." Confidence intervals should be provided for these differences, especially given the small PDX cohort size which may have substantial sampling error.

Response: We fully agree with the reviewer that reporting simple percentage point differences without statistical context may be insufficient, especially given the relatively small size of the PDX cohort. To address this concern, we have revised the manuscript to include Chi-square tests and Fisher’s exact tests, as appropriate, to assess the statistical significance of the observed differences in mutation frequencies. Where applicable, Chi-square values are now reported to provide further analytical clarity.

  1. "However, while the clinical cohort demonstrates infrequent instances of concomitant mutations (e.g., KRAS + BRAF, NRAS + ERBB2), no co-occurrence of these mutations was observed in the PDX cohort." For known mutually exclusive mutation pairs (e.g., KRAS and BRAF), specific mutual exclusivity analysis should be performed, with calculation of exclusivity coefficients, rather than just descriptive reporting.

Response: Thank you for the valuable suggestion. In response, we have conducted a statistical analysis of mutual exclusivity and co-occurrence among the six major mutation types in CRC. This analysis included the calculation of exclusivity coefficients to quantitatively assess the likelihood of mutations being mutually exclusive versus co-occurring. The results provide a more robust understanding of the mutational relationships within the cohort and will be included in the revised manuscript.

  1. KRAS mutation frequencies reported in section 2.3 (40.2% and 20.1%) differ slightly from those in section 2.1 (40.9% and 20.1%) without explanation.

Response: Since there were novel data in the public databases, the analysis on the clinical cohort was updated and re-done. We reviewed all the novel numerical data and eliminated any inconsistencies found in the previous version of the manuscript. The revised manuscript now reflects up-to-date, accurate and consistent information. Consequently, any potential discrepancy in KRAS mutation frequencies has been resolved, too. The values are now consistent across all sections and based on the same dataset and filtering criteria.           

  1. "The most plausible explanation for these observed differences is selective engraftment bias, which is the preferential engraftment of more aggressive tumors with stronger growth potential [16]." Discussion attributes BRAF mutation enrichment simply to "selective engraftment bias" without exploring how BRAF mutations specifically affect cellular behavior to promote PDX establishment.

Response: We thank the reviewer for this insightful comment. In response, we have expanded the discussion to clarify that BRAF mutations confer a constitutive activation of the MAPK/ERK pathway, enabling tumor cells to proliferate and survive independently of external growth signals. This intrinsic signaling advantage allows BRAF-mutant cells to better withstand the stresses of transplantation—such as the lack of human stromal support and growth factors—and promotes autonomous adaptation within the murine host. Additionally, these tumors often upregulate pro-angiogenic factors like VEGF, facilitating rapid vascularization and enhancing their ability to establish and grow in PDX models.

  1. "Furthermore, successive passages may result in the loss of subclonal heterogeneity in PDX models." The paper mentions passage issues multiple times but doesn't specify which passage of PDX models was used in this study, nor discuss how passage number specifically affected these results, making this discussion theoretical rather than practical.

Response: We now specify that all samples used were from the very early passages 0–3. We also updated the Discussion as well as the Material and Methods sections to reflect how early-passage models aim to preserve original tumor heterogeneity while minimizing clonal drift, thus giving more practical context to the limitations discussed.

Round 2

Reviewer 1 Report

Comments and Suggestions for Authors

The authors have adequately responded to all suggestions

Reviewer 3 Report

Comments and Suggestions for Authors

Dear Authors,

The addition of methodological details, statistical analyses, and the mechanistic explanation for BRAF mutation enrichment have improved the manuscript significantly.

The numerical inconsistencies have been resolved and the justification for your analytical approach is now clear.